# IMPROVING DIFFERENTIALLY PRIVATE MODELS WITH ACTIVE LEARNING

## ABSTRACT

Broad adoption of machine learning techniques has increased privacy concerns for models trained on sensitive data such as medical records. Existing techniques for training differentially private (DP) models give rigorous privacy guarantees, but applying these techniques to neural networks can severely degrade model performance. This performance reduction is an obstacle to deploying private models in the real world. In this work, we improve the performance of DP models by fine-tuning them through active learning on public data. We introduce two new techniques — DIVERSEPUBLIC and NEARPRIVATE — for doing this fine-tuning in a privacy-aware way. For the MNIST and SVHN datasets, these techniques improve state-of-the-art accuracy for DP models while retaining privacy guarantees.

## 1 INTRODUCTION

Privacy concerns surfaced with the increased adoption of machine learning in domains like healthcare (LeCun et al., 2015). One widely adopted framework for measuring privacy characteristics of randomized algorithms, such as machine learning techniques, is differential privacy (Dwork et al., 2006). Abadi et al. (2016) introduced an algorithm for differentially private stochastic gradient descent (DP-SGD), which made it feasible to scale differential privacy guarantees to neural networks. DP-SGD is now the de facto algorithm used for training neural networks with privacy guarantees.

There is, however, a crucial problem for models with a large number of parameters: it is difficult to achieve both non-trivial privacy guarantees and good accuracy. The reasons for this are numerous and involved, but the basic intuition is that DP-SGD involves clipping gradients and then adding noise to those gradients. However, gradient clipping has an increasing relative effect as the number of model parameters grows. This reduces the applicability of differential privacy to deep learning in practice, where strong performance tends to require a large number of model parameters.[1]

In this work, we propose a method to mitigate this problem, making DP learning feasible for modern image classifiers. The method is based on the observation that each DP-SGD update consumes privacy budget in proportion to the quotient of the batch size and the training-set size. Thus, by increasing the number of effective training examples, we can improve accuracy while maintaining the same privacy guarantees. Since obtaining more private training samples will generally be nontrivial, we focus on using 'public' data to augment the training set. This involves assuming that there will be unlabeled[2] public data that is sufficiently related to our private data to be useful, but we think this is a reasonable assumption to make, for two reasons: First, it is an assumption made by most of the semi-supervised learning literature (Oliver et al., 2018). Second, one of our techniques will explicitly address the situation where the 'sufficiently related' clause partially breaks down.

In summary, our contributions are:

- We introduce an algorithm (DIVERSEPUBLIC ) to select diverse representative samples from a public dataset to fine-tune a DP classifier.

---

[1]For example, the best performing CIFAR-10 classifier from Zagoruyko & Komodakis (2016) has 32.5 million parameters, while the private baseline we describe in Section 4.1 has about 26 thousand.

[2] If there is labeled public data, the situation is even better, but this is likely to be rare and is not the setting we consider here.

- We then describe another algorithm (NEARPRIVATE ) that pays a privacy cost to reference the private training data when querying the public dataset for representative samples.
- We establish new state-of-the-art results on the MNIST and SVHN datasets by fine-tuning DP models using simple active learning techniques, and then improve upon those results further using DIVERSEPUBLIC and NEARPRIVATE .
- We open source all of our experimental code.[3]

## 2 BACKGROUND

### 2.1 DIFFERENTIAL PRIVACY

We reason about privacy within the framework of differential privacy (Dwork et al., 2006). In this paper, the random algorithm we analyze is the training algorithm, and guarantees are measured with respect to the training data.

Informally, an algorithm is said to be differentially private if its behavior is indistinguishable on pairs of datasets that only differ by one point. That is, an observer cannot tell whether a particular point was included in the model's training set simply by observing the output of the training algorithm. Formally, for a training algorithm $A$ to be $(\varepsilon, \delta)$-differentially private, we require that it satisfies the following property for all pairs of datasets $(d, d')$ differing in exactly one data point and all possible subsets $S \subset \text{Range}(A)$:

$$Pr[A(d) \in S] \leq e^{\varepsilon} Pr[A(d') \in S] + \delta$$

where $\delta$ is a (small) probability for which we tolerate the property not to be satisfied. The parameter $\varepsilon$ measures the strength of the privacy guarantee: the smaller the value of $\varepsilon$ is, the stronger the privacy guarantee is.

This guarantee is such that the output of a differentially private algorithm can be post-processed at no impact to the strength of the guarantee provided. In case privacy needs to be defined at a different granularity than invidividual training points, the guarantee degrades by a factor which, naively, is the number of points that are included in the granularity considered. See Dwork et al. (2014a) for further information.

### 2.2 DIFFERENTIALLY PRIVATE STOCHASTIC GRADIENT DESCENT (DP-SGD)

Building on earlier work (Chaudhuri et al., 2011; Song et al., 2013), Abadi et al. (2016) introduce a variant of stochastic gradient descent to train deep neural networks with differential privacy guarantees. Two modifications are made to vanilla SGD. First, gradients are computed on a per-example basis and clipped to have a maximum known $\ell_2$ norm. This bounds the sensitivity of the training procedure with respect to each individual training point. Second, noise calibrated to have a standard deviation proportional to this sensitivity is added to the average gradient. This results in a training algorithm known as differentially private SGD (DP-SGD). Unfortunately (as discussed in Section 1), DP-SGD does not perform well for models with large parameter counts, which motivates the improvements proposed in the next section.

## 3 IMPROVING DIFFERENTIALLY PRIVATE MODELS WITH ACTIVE LEARNING

This paper introduces the following high-level process to improve the performance of an existing DP classifier: First, find a public insensitive dataset containing relevant samples. Second, carefully select a subset of public samples that can improve the performance of the classifier. Third, fine-tune the existing classifier on the selected samples with labels. We want to perform the selection and the fine-tuning in a way which does not compromise the privacy guarantees of the existing classifier.

The first step can be done using domain knowledge about the classifier, e.g., utilizing relevant public genomic datasets for a DP classifier of genomics data. We assume standard fine-tuning techniques for the last step. Therefore, the problem boils down to efficiently selecting samples from the public dataset while preserving privacy. We also assume a limit on the number of selected samples. This

---

[3]URL blinded for anonymity.

---

**Algorithm 1** The DIVERSEPUBLIC Algorithm

---

**Input:** $\mathcal{M}_{dp}$ with privacy cost $\varepsilon_{dp}$, $\mathcal{D}_{public}$, $N_{labeled}$, $k > N_{labeled}$, $p$, $N_{cluster}$, $N_{each}$
**Output:** $\mathcal{M}_{dp}^*$ fine-tuned on selected public data with the same privacy cost $\varepsilon_{dp}$

    $E_{public} \leftarrow \mathcal{M}_{dp}(\mathcal{D}_{public})[-1]$               ▷ Compute 'embeddings' of public data
    $P \leftarrow \text{PCA}(E_{public}, p)$           ▷ Perform PCA on embeddings and get first $p$ PCs
    $S_{public} \leftarrow P \times \text{SelectUncertain}(\mathcal{D}_{public}, k)$     ▷ Project $k$ most 'uncertain' public data onto PCs
    $\text{ClusterCenters} \leftarrow \text{K-Means}(S_{public})$     ▷ Cluster projected embeddings to get $N_{cluster}$ clusters
    $\mathcal{D}_{labeled} \leftarrow \emptyset$
    **for** $i = 1$ to $N_{cluster}$ **do**           ▷ Label $N_{each}$ data points from each cluster
        $\mathcal{D}_{labeled} = \mathcal{D}_{labeled} \cup \text{TakeCenterPoints}(\text{ClusterCenters})$
    **end for**
    $\mathcal{M}_{dp}^* \leftarrow \text{FineTune}(\mathcal{M}_{dp}, \mathcal{D}_{labeled})$

---

limit is relevant when the samples are unlabeled, in which case it controls the cost of labeling (e.g., hiring human annotators to process the selected samples).

We introduce two active learning algorithms, DIVERSEPUBLIC and NEARPRIVATE , for sample-selection. These algorithms make different assumptions about access to the private training data but are otherwise drop-in replacements in the end-to-end process.

### 3.1 PROBLEM STATEMENT AND BASELINE METHODS

We are given a differentially private model $\mathcal{M}_{dp}$ trained and tested on private sensitive data $\mathcal{D}_{train}$ and $\mathcal{D}_{test}$ respectively. The privacy cost of training $\mathcal{M}_{dp}$ on $\mathcal{D}_{train}$ is $\varepsilon_{dp}$ (we omit $\delta$ for brevity in this paper, but it composes similarly). And we have an extra set of public insensitive unlabeled data $\mathcal{D}_{public}$ which can be utilized to further improve $\mathcal{M}_{dp}$. Given the number of extra data $N_{labeled}$ that we are allowed to request labels for, and the total privacy budget $\varepsilon_{limit}$ of the improved model $\mathcal{M}_{dp}^*$, we want to efficiently pick $\mathcal{D}_{labeled} \subset \mathcal{D}_{public}$ where $|\mathcal{D}_{labeled}| = N_{labeled}$, using which we can fine-tune $\mathcal{M}_{dp}$ to $\mathcal{M}_{dp}^*$ of better performance on $\mathcal{D}_{test}$. The simplest baseline method we will use is to choose $N_{labeled}$ 'random' samples out of $\mathcal{D}_{public}$ for fine-tuning. The other baseline that we use is Uncertain Sampling (Settles, 2009), according to either the 'entropy' of the logits or the 'difference' between the two largest logits. These are widely used active learning methods and serve as strong baselines.

### 3.2 THE DIVERSEPUBLIC METHOD

The baseline methods may select many redundant examples to label. To efficiently select diverse representative samples, we propose the DIVERSEPUBLIC method, adapted from clustering based active learning methods (Nguyen & Smeulders, 2004). Given a DP model $\mathcal{M}_{dp}$ of cost $\varepsilon_{dp}$, we obtain the 'embeddings' (the activations before the logits) $E_{public}$ of all $\mathcal{D}_{public}$, and perform PCA on $E_{public}$ as is done in Raghu et al. (2017). Then we select a number (more than $N_{labeled}$) of uncertain points (according to, e.g., the logit entropy of the private model) from $\mathcal{D}_{public}$, project their embeddings onto the top few principal components, and cluster those projections into groups. Finally, we pick a number of samples from each representative group up to $N_{labeled}$ in total and fine-tune the DP model with these $\mathcal{D}_{labeled}$. Though this procedure accesses $\mathcal{M}_{dp}$, it adds nothing to $\varepsilon_{dp}$, since there is no private data referenced and the output of DP models can be post-processed at no additional privacy cost. *It can be applied even to models for which we cannot access the original training data.*[4] Algorithm 1 presents more details of the DIVERSEPUBLIC method.

### 3.3 THE NEARPRIVATE METHOD

The DIVERSEPUBLIC method works well under the assumption that $\mathcal{D}_{public}$ has the same distribution as $\mathcal{D}_{train}$. However, this may not be a reasonable assumption in general (Oliver et al., 2018). For instance, there may be a subset of $\mathcal{D}_{public}$ about which our pre-trained model has high uncertainty but which cannot improve performance if sampled. This may be because that subset contains corrupted data or it may be due simply to distribution shift. In order to mitigate this issue, we propose to

---

[4]e.g., models published to a repository like TensorFlow Hub: https://www.tensorflow.org/hub

---

**Algorithm 2** The NEARPRIVATE Algorithm

---

**Input:** $\mathcal{M}_{\text{dp}}$ with privacy cost $\varepsilon_{\text{dp}}, \mathcal{D}_{\text{public}}, N_{\text{labeled}}, k > N_{\text{labeled}}, p, \mathcal{D}_{\text{train}}, \varepsilon_{\text{dpPCA}}$ and $\varepsilon_{\text{support}}$
**Output:** $\mathcal{M}_{\text{dp}}^{*}$ fine-tuned on selected public data with privacy cost $\varepsilon_{\text{dp}} + \varepsilon_{\text{dpPCA}} + \varepsilon_{\text{support}}$

    $E_{\text{train}} \leftarrow \mathcal{M}_{\text{dp}}(\mathcal{D}_{\text{train}})[-1]$               ▷ Compute 'embeddings' of private data
    $P \leftarrow \text{DP-PCA}(E_{\text{train}}, \varepsilon_{\text{dpPCA}}, p)$      ▷ Perform DP-PCA on embeddings and get first $p$ PCs
    $S_{\text{train}} \leftarrow P \times \text{SelectUncertain}(\mathcal{D}_{\text{train}}, k)$     ▷ Project $k$ most 'uncertain' private data onto PCs
    $S_{\text{public}} \leftarrow P \times \text{SelectUncertain}(\mathcal{D}_{\text{public}}, k)$     ▷ Project $k$ most 'uncertain' public data onto PCs
    $\text{NeighborCounts} \leftarrow \emptyset$
    **for** $a \in S_{\text{train}}$ **do**            ▷ In PC-space, assign each private point to exactly one public point
        $\text{Increment}(\text{NeighborCounts}, \text{NearestNeighbor}(a, S_{\text{public}}))$
    **end for**
    **for** $b \in S_{\text{public}}$ **do**          ▷ Compute support with Laplacian noise for each public data point
        $N_{\text{support}}(b) \leftarrow \text{NeighborCounts}(b) + \text{LaplaceNoise}(1/\varepsilon_{\text{support}})$
    **end for**
    $\mathcal{D}_{\text{labeled}} \leftarrow \text{TakeArgmaxPoints}(N_{\text{support}})$     ▷ Label $N_{\text{labeled}}$ data points of the highest $N_{\text{support}}$
    $\mathcal{M}_{\text{dp}}^{*} \leftarrow \text{FineTune}(\mathcal{M}_{\text{dp}}, \mathcal{D}_{\text{labeled}})$

---

check query points against our private data and decline to label query points that are too far from any training points (in projected-embedding-space). But doing this check while preserving privacy guarantees is nontrivial, since it involves processing the private training data itself in addition to $\mathcal{M}_{\text{dp}}$. Given a DP model of cost $\varepsilon_{\text{dp}}$, we obtain the embeddings $E_{\text{train}}$ of all $\mathcal{D}_{\text{train}}$, and perform differentially private PCA (DP-PCA) (Dwork et al., 2014b) on $E_{\text{train}}$ at a privacy cost of $\varepsilon_{\text{dpPCA}}$. We select a number of uncertain points from both $\mathcal{D}_{\text{train}}$ and $\mathcal{D}_{\text{public}}$. Then in the space of low dimensional DP-PCA projections, we assign each uncertain private example to exactly one uncertain public example according to Euclidean distance. This yields, for each uncertain public example, a count $N_{\text{support}}$ of 'nearby' uncertain private examples. Finally, we choose $N_{\text{labeled}}$ points with the largest values of $N_{\text{support}} + \text{LaplaceNoise}(1/\varepsilon_{\text{support}})$ from the uncertain public data. This sampling procedure is differentially private with cost $\varepsilon_{\text{support}}$ due to the Laplace mechanism (Dwork et al., 2014a). The total privacy cost of NEARPRIVATE is composed of the budgets expended to perform each of the three operations that depend on the private data: $\varepsilon_{\text{dp}} + \varepsilon_{\text{dpPCA}} + \varepsilon_{\text{support}}$. More details in Algorithm 2.

## 4 EXPERIMENTS

We conduct experiments on the MNIST (LeCun et al., 1998) and SVHN (Netzer et al., 2011) data sets. These may be seen as 'toy' data sets in the object recognition literature, but they are still challenging for DP object recognizers. In fact, at the time of this writing, there are no published examples of a differentially private SVHN classifier with both reasonable accuracy and non-trivial privacy guarantees. *The baseline we establish in Section 4.2 is thus a substantial contribution by itself.* For both datasets, we use the same model architecture as in the Tensorflow Privacy tutorials[5]. We obtain $\mathcal{M}_{\text{dp}}$ by training on $\mathcal{D}_{\text{train}}$ via DP-SGD (Abadi et al., 2016). Unless otherwise specified, we always aggregate results over 5 runs with different random seeds and use error bars to represent the standard deviation. We use the implementation of DP-SGD made available through the TensorFlow Privacy library (McMahan et al., 2018) with $\delta = 10^{-5}$.

### 4.1 EXPERIMENTS ON MNIST

We conduct our first set of experiments on the MNIST (LeCun et al., 1998) dataset. We use the Q-MNIST (Yadav & Bottou, 2019) dataset as our source of public data. In particular, we use the 50,000 examples from the Q-MNIST dataset that are reconstructions of the lost MNIST testing digits. We perform a hyperparameter optimization and find a baseline model with higher accuracy (97.3% at $\varepsilon_{\text{dp}} = 3.0$, and 97.5% at $\varepsilon_{\text{dp}} = 3.2$) than what is reported as the current state-of-the-art in the Tensorflow Privacy tutorial's README file (96.6% at $\varepsilon_{\text{dp}} = 3.0$).

Figure 1a shows results of our DIVERSEPUBLIC method compared with baselines. Starting from a checkpoint of test accuracy 97.5% ($\varepsilon_{\text{dp}} = 3.2$), our method can reach 98.8% accuracy with 7,000

---

[5]Tutorials for TensorFlow Privacy are found at: `https://github.com/tensorflow/privacy`

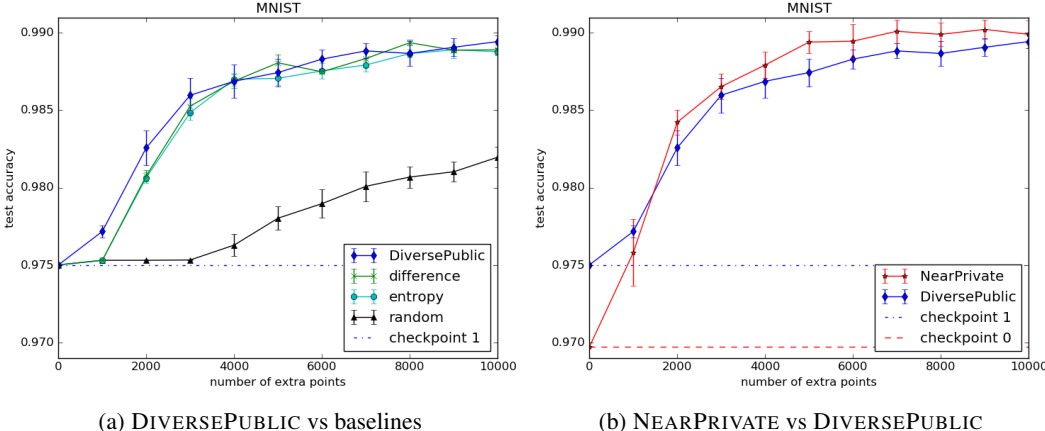

(a) DIVERSEPUBLIC vs baselines

(b) NEARPRIVATE vs DIVERSEPUBLIC

Figure 1: **MNIST experiments.** All plots show test-set-accuracy vs. the number of extra points labeled. **Left:** Comparison of our DIVERSEPUBLIC technique with the active learning baselines described above. All models are fine-tuned starting from the same baseline ('checkpoint 1': test accuracy 97.5% and $\varepsilon_{dp} = 3.2$). Active learning improves the performance of the DP-model to as much as 98.8% in the best case with no increase in privacy cost. **Right:** Comparison of our NEARPRIVATE technique with the DIVERSEPUBLIC technique. Since we spent $\varepsilon_{dpPCA} + \varepsilon_{support} = 1.0$ on the NEARPRIVATE technique, we fine-tune the NEARPRIVATE model from 'checkpoint 0' with privacy cost $\varepsilon_{dp} = 2.2$. Thus, both lines have the same privacy cost ($\varepsilon_{limit} = 3.2$), regardless of the number of extra points used.

extra labels. The DIVERSEPUBLIC method yields higher test accuracy than other active learning baselines in the low-query regime, and performs comparably in the high-query regime.

In Figure 1b, we compare NEARPRIVATE against DIVERSEPUBLIC . Since NEARPRIVATE adds extra privacy cost, we have to take special care when comparing it to DIVERSEPUBLIC . Therefore, we fine-tune from a starting checkpoint at test accuracy 97.0% with lower privacy cost ($\varepsilon_{dp} = 2.2$) and make sure its total privacy cost ($\varepsilon_{limit} = 3.2$) in the end is the same as the cost for the DIVERSEPUBLIC model. For this reason, NEARPRIVATE takes some number of labeled data points to 'catch up' to DIVERSEPUBLIC for the same DP cost — in this case 2,000. When $N_{labeled}$ is large enough, NEARPRIVATE outperforms all other methods. This shows that accessing the original training data in a privacy-aware way can substantially improve performance.

## 4.2 EXPERIMENTS ON SVHN

We conduct another set of experiments on the SVHN (Netzer et al., 2011) data. We use the set of '531,131 additional, somewhat less difficult samples' as our source of public data. Since a baseline model trained with DP-SGD on the SVHN training set performs quite poorly, we have opted to first pre-train the model on rotated images of $\mathcal{D}_{public}$ predicting only rotations as in Gidaris et al. (2018).

Broadly speaking, the results presented in Figure 2 are similar to MNIST results, but three differences stand out. First, the improvement given by active learning over the baseline private model is larger. Second, the improvement given by DIVERSEPUBLIC over the basic active learning techniques is also larger. Third, NEARPRIVATE actually underperforms DIVERSEPUBLIC in this case. We hypothesize that the first and second results are due to there being more 'headroom' in SVHN accuracy than for MNIST, and that the third result stems from the reported lower difficulty of the extra SVHN data. In the next section, we examine this phenomenon further.

## 4.3 EXPERIMENTS WITH DATASET POLLUTION

We were intrigued by the under-performance of NEARPRIVATE relative to DIVERSEPUBLIC on SVHN. We wondered whether it was due to the fact that SVHN and its extra data violate the assumption built into NEARPRIVATE – namely that we need to query the private data to 'throw out' unhelpful public data. Indeed, the SVHN website describes the extra set as 'somewhat less

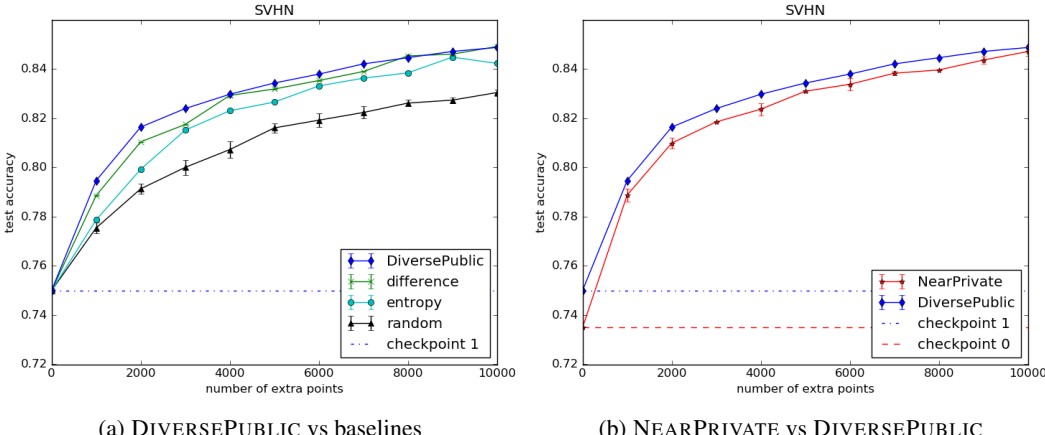

(a) DIVERSEPUBLIC vs baselines  (b) NEARPRIVATE vs DIVERSEPUBLIC

Figure 2: **SVHN experiments. Left:** Comparison of our DIVERSEPUBLIC technique with the active learning baselines. All models are fine-tuned starting from the same baseline ('checkpoint 1': test accuracy 75.0% and $\varepsilon_{dp} = 6.0$). Active learning improves the performance of the DP-model to as much as 85% in the best case with no increase in privacy cost. Recall also that the 75.0% number is itself a baseline established in this paper. **Right:** Comparison of NEARPRIVATE with DIVERSEPUBLIC . The setup here is analogous to the one in the MNIST experiment, but DIVERSEPUBLIC performs better in this case. Since we spent $\varepsilon_{dpPCA} + \varepsilon_{support} = 1.0$ on NEARPRIVATE , we fine-tune from 'checkpoint 0' with test accuracy 73.5% and $\varepsilon_{dp} = 5.0$.

difficult' than the training data. To test this hypothesis, we designed a new experiment to check if NEARPRIVATE can actually select more helpful samples given a mixture of relevant data and irrelevant data as 'pollution'. We train the DP baseline with 30,000 of the SVHN training images, and treat a combination of another 40,000 SVHN training images and 10,000 CIFAR-10 (Krizhevsky et al., 2009) training images as the extra public dataset. These CIFAR-10 examples act as the unhelpful public data that we would hope NEARPRIVATE could learn to discard. As shown in Figure 3, all baselines perform worse than before with polluted public data. DIVERSEPUBLIC does somewhat better than random selection, but not much, achieving a peak performance improvement of around 1%. On the other hand, the difference between NEARPRIVATE and DIVERSEPUBLIC is more substantial, at over 2% accuracy in some cases. This is especially interesting considering that DIVERSEPUBLIC actually performed better in the results of Section 4.2. Broadly speaking, the results support our claim that NEARPRIVATE helps more relative to DIVERSEPUBLIC when there is 'unhelpful' data in the public dataset. This is good to know, since having some unhelpful public data and some helpful public data seems like a more realistic problem setting than the one in which all public data is useful.

## 5 ABLATION ANALYSES

In order to better understand how the performance of DIVERSEPUBLIC and NEARPRIVATE is affected by various hyper-parameters, we conduct several ablation studies.

### 5.1 HOW DO CLUSTERING HYPER-PARAMETERS AFFECT ACCURACY?

For DIVERSEPUBLIC , there are two parameters that affect the number of extra data points labeled for fine-tuning: the number of clusters we form ($N_{cluster}$) and the number of points we label per cluster ($N_{each}$). We write $N_{labeled} = N_{cluster} \times N_{each}$. To study the relative effects of these, we conduct the experiment depicted in Figure 4a. In this experiment, we fine-tune the same DP MNIST model (test accuracy 97% at $\varepsilon_{dp} = 2.2$) with varying values of $N_{cluster}$ and $N_{each}$. We vary $N_{cluster}$ from 100 to 500, which is depicted on the $x$-axis. We vary $N_{each}$ from 5 to 20, with each value depicted as a different line. The general trend is one of diminishing returns on extra labeled data, as would be predicted by Hestness et al. (2017). We do not notice a strong correspondence between final test accuracy at a fixed number of extra labels and the values of $N_{cluster}$ and $N_{each}$. This is encouraging, as it suggests that practitioners can use our techniques without worrying too much about these hyper-parameters.

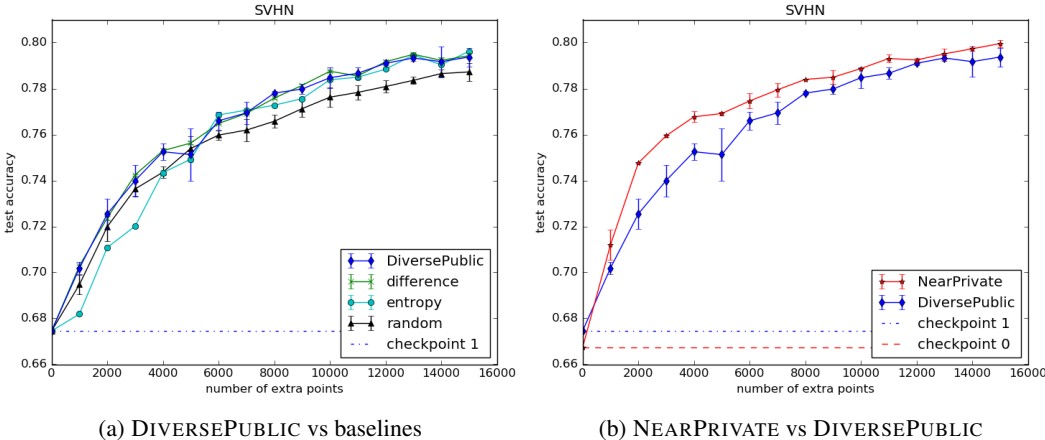

(a) DIVERSEPUBLIC vs baselines

(b) NEARPRIVATE vs DIVERSEPUBLIC

Figure 3: **SVHN experiments with dataset pollution.** In this experiment, we train the DP baseline with 30,000 of the SVHN training images. The extra public dataset is a combination of 40,000 SVHN training images and 10,000 CIFAR-10 training images. **Left:** DIVERSEPUBLIC compared against other active learning baseline techniques as in Figure 2a. In this case, the active learning techniques do not outperform random selection by very much. Uncertainty by itself is not a sufficient predictor of whether extra data will be helpful here, since the baseline model is also uncertain about the CIFAR-10 images. **Right:** NEARPRIVATE compared against DIVERSEPUBLIC , but started from different checkpoints ('checkpoint 0' with $\varepsilon_{dp} = 5.0$, 'checkpoint 1' with $\varepsilon_{dp} = 6.0$) to keep the privacy cost constant as in Figure 2b. In this case, NEARPRIVATE substantially outperforms DIVERSEPUBLIC by selecting less of the CIFAR-10 images.

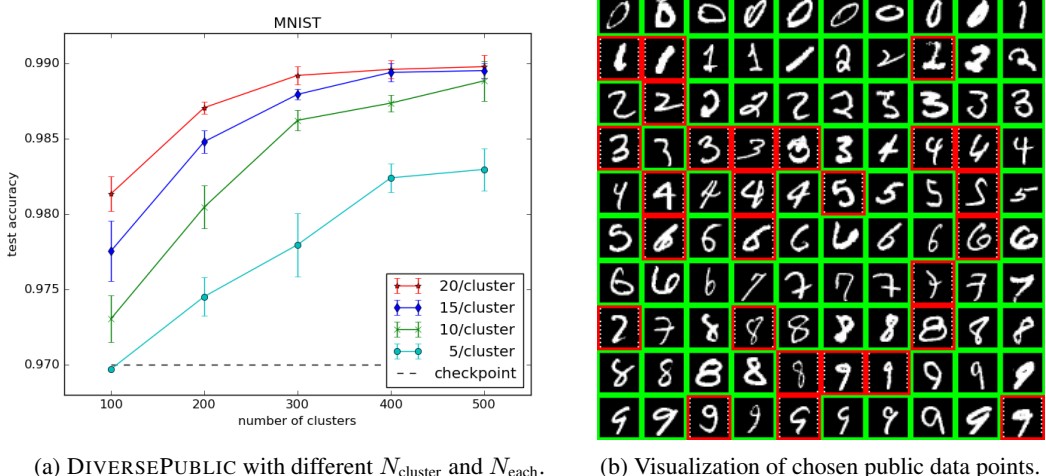

(a) DIVERSEPUBLIC with different $N_{cluster}$ and $N_{each}$.

(b) Visualization of chosen public data points.

Figure 4: DIVERSEPUBLIC **analysis. Left:** We apply DIVERSEPUBLIC to the same DP checkpoint (dashed horizontal line), varying the number of clusters (horizontal axis) and the number of chosen points from each cluster (lines with error bars). **Right:** For $N_{cluster} = 100, N_{each} = 20$, we visualize the most central example from each cluster. Since NEARPRIVATE does not have explicit clustering, we use DIVERSEPUBLIC for this visualization. Green borders mean that the initial checkpoint (dashed horizontal line) predicted correctly; while red bordered examples (originally predicted incorrectly) have dots on the left showing predictions and dots on the right showing true labels.

To address the question of which extra data points are being chosen for labeling, we create Figure 4b showing the most central example from each cluster, computed using $N_{cluster} = 100, N_{each} = 20$. The chosen examples are quite diverse, with a similar number of representatives from each class and variations in the thickness and shear of the digits. We can also inspect examples labeled incorrectly by the original checkpoint, such as the digit in Row 8, Col 1, which is a 7 that looks a lot like a '2'.

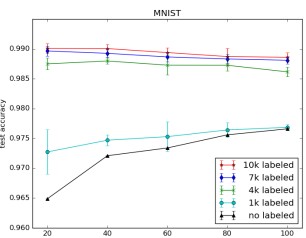 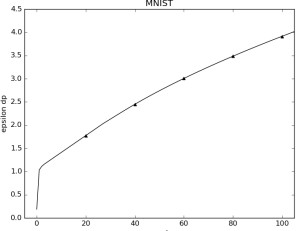 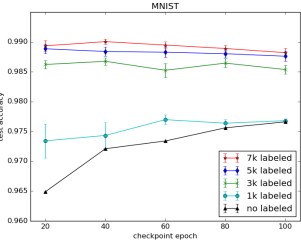

(a) Fixed extra privacy cost with $\varepsilon_{\text{dpPCA}} = 0.5$, $\varepsilon_{\text{support}} = 0.5$.

(b) The initial training privacy cost $\varepsilon_{\text{dp}}$ for different checkpoints.

(c) Fixed total privacy cost with $\varepsilon_{\text{limit}} = 5.0$, $\varepsilon_{\text{dpPCA}} = 0.3$.

Figure 5: NEARPRIVATE **analysis.** We apply NEARPRIVATE to DP checkpoints (horizontal axis) of different privacy cost $\varepsilon_{\text{dp}}$. The black line with triangle represents those starting checkpoints. **Middle:** The initial training privacy cost $\varepsilon_{\text{dp}}$ of DP checkpoints at different epochs of a single training run. **Left:** We fix the extra privacy cost $\varepsilon_{\text{dpPCA}} + \varepsilon_{\text{support}} = 1.0$, but vary the number of labeled public points for each line. We achieve large improvements from 1,000 to 4,000 labeled public points. With even larger $N_{\text{labeled}}$, the improvements are not significant and it may be better to start fine-tuning from a DP checkpoint of lower privacy cost. **Right:** With $\varepsilon_{\text{dpPCA}}$ set to 0.3, we fix the total privacy cost $\varepsilon_{\text{limit}} = 5.0$ and vary $\varepsilon_{\text{support}}$. For $N_{\text{labeled}} = 1000$, fine-tuning from the checkpoint at Epoch 60 is the best given a total privacy budget of 5.0.

## 5.2 HOW DOES THE STARTING CHECKPOINT AFFECT RESULTS?

Recall that NEARPRIVATE accrues extra privacy cost by accessing the histogram of neighbor counts. This means that achieving a given accuracy under a constraint on the total privacy cost requires choosing how to allocate privacy between NEARPRIVATE and the initial DP-SGD procedure. Making this choice correctly requires a sense of how much benefit can be achieved from applying NEARPRIVATE to different starting checkpoints. Toward that end, we conduct (Figure 5) an ablation experiment on MNIST where we run NEARPRIVATE on many different checkpoints from the same training run. Figure 5a shows the test accuracies resulting from fine-tuning checkpoints at different epochs (represented on the $x$-axis) with fixed extra privacy cost of $1.0$. Figure 5b shows the corresponding $\varepsilon_{\text{dp}}$ for each checkpoint. Figure 5c varies other parameters given a fixed total privacy budget of $5.0$.

In Figure 5a, the black line with triangle markers shows the initial test accuracies of the checkpoints. The other lines show results with different values of $N_{\text{labeled}}$ from 1,000 to 10,000 respectively. With $N_{\text{labeled}} = 1000$, improvements are marginal for later checkpoints. In fact, the improvement from using $N_{\text{labeled}} = 1000$ at checkpoint 80 is not enough to compensate for the additional 1.0 privacy cost spent by NEARPRIVATE , because you could have had the same increase in accuracy by training the original model for 20 more epochs, which costs less than that. On the other hand, the improvements from using $N_{\text{labeled}} = 4000$ or higher are significant and cannot be mimicked by training for longer.

Given a total privacy budget $\varepsilon_{\text{limit}}$, how should we decide among $\varepsilon_{\text{dp}}$, $\varepsilon_{\text{dpPCA}}$, and $\varepsilon_{\text{support}}$? Empirically, we observe that $\varepsilon_{\text{dpPCA}}$ can be set to a small value (e.g., 0.3) without substantially affecting the results. Allocating between $\varepsilon_{\text{dp}}$ and $\varepsilon_{\text{support}}$ is addressed in Figure 5c, which varies those parameters with $\varepsilon_{\text{limit}}$ fixed to 5.0. When the budget for gathering $\mathcal{D}_{\text{labeled}}$ is low, say $N_{\text{labeled}}$ is around 1000, it is preferable to pick a later DP checkpoint, consuming a higher $\varepsilon_{\text{dp}}$ and lower $\varepsilon_{\text{support}}$. On the other hand, when allowed to label more instances from the public data, we should use an earlier DP checkpoint (with a lower $\varepsilon_{\text{dp}}$) and choose better public samples with respect to the private data.

## 6 CONCLUSION

In addition to creating new baselines for DP image classifiers by fine-tuning on public data, we introduce two algorithms – DIVERSEPUBLIC and NEARPRIVATE – to perform fine-tuning in a privacy-aware way. We conduct experiments showing that these algorithms bring DP object recognition closer to practicality, improving on the aforementioned benchmarks. We hope that this work will encourage further research into techniques for making differential privacy more useful in practice, and we hope that the techniques we propose here will be helpful to existing practitioners.

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
