# OpenReview forum: "Improving Differentially Private Models with Active Learning"
_ICLR.cc/2020/Conference — Reject_

### Official Review · AnonReviewer2 · 2019-10-23
**Official Blind Review #2**

**Rating:** 1

**Review:**

The paper proposes methods for improving the performance of differentially private models when additional related but public data is available. The authors propose to fine-tune the private model with related but public data. Two heuristics for selecting data to fine-tune with are presented.

I vote for rejecting the paper. My main concerns revolve around the paper's contributions. It neither makes core contributions to differential privacy nor to active learning. What it does instead is present a post-processing procedure which boils down to fine-tuning differentially private models with public data. Arguably the main contribution of the paper is the heuristics for selecting public instances for fine-tuning. These rely on several design choices (why k-means? how are points from within a cluster sampled -- uniformly at random, some other heuristic? how are uncertain private and public points matched, by solving an assignment problem or something else? ) that are not well justified or have missing details and more importantly do not clearly outperform baseline acquisition functions - selecting points based on predictive entropy or the difference between two largest logits. Finally, I am unconvinced that the paper’s premise of having access to related public data is realistic.

This would likely not change my opinion of the paper, but it would be good to substantiate the claims made in the second paragraph of the paper — the performance of differentially private models degrades with increasing model size. If the relationship is as clear as the paragraph makes it sound, then it would be great to include a graph with the x-axis being the number of parameters and the y-axis being the difference in performance between a regular and a differentially private model. One would expect to see the difference increase with model size.

**Experience Assessment:**

I have read many papers in this area.

**Review Assessment: Checking Correctness Of Derivations And Theory:**

N/A

**Review Assessment: Checking Correctness Of Experiments:**

I assessed the sensibility of the experiments.

**Review Assessment: Thoroughness In Paper Reading:**

I read the paper thoroughly.

---

> ### Author Response · Authors · 2019-11-15
> **Hi**
>
> Hi, responses inline, for posterity:
>
> > It neither makes core contributions to differential privacy nor to active learning.
> See our response to Reviewer 4 on this point.
>
> > Arguably the main contribution of the paper is the heuristics for selecting public instances for fine-tuning.
> We would argue that the main contribution is showing that simple techniques can substantially improve the
> state of the art for differentially private classifiers, but again, see our response to reviewer 4.
>
> > do not clearly outperform baseline acquisition functions
> First, figures 1 and 2 clearly show that they do outperform the baseline acquisition functions,
> so we're slightly confused by this comment?
> Second, and more importantly, the baselines are (again see our response to R4) baselines we establish in this paper.
> There was no prior literature on using active learning for this purpose,
> and we claim that establishing those baselines is one of the main contributions of this paper.
>
> > Finally, I am unconvinced that the paper’s premise of having access to related public data is realistic.
> This is the main premise of [1], one of the most highly-cited papers in this subfield and an Oral presentation
> at ICLR 2017.
>
> [1] "Semi-supervised Knowledge Transfer for Deep Learning from Private Training Data". Papernot et al. 2017.
> (  https://openreview.net/forum?id=HkwoSDPgg  )

---

### Official Review · AnonReviewer4 · 2019-10-27
**Official Blind Review #4**

**Rating:** 3

**Review:**

---SUMMARY---
This paper gives two methods for improving an existing differentially private model. Both methods assume a source of unlabeled data that requires no privacy guarantees. The two methods make different uses of this public data to augment the original private model. Both preserve (a degree of) privacy for the original model.

The first method is DiversePublic. DiversePublic only post-processes the original private model M -- in particular, it does not touch the original data -- and so its privacy is immediate. At a high level, DiversePublic works by picking the "best" unlabelled data points for augmenting M. First, it applies M to the unlabeled data and then applies PCA to the result. Next, it selects highly "uncertain" points, projects them onto the top k principal components, and clusters the results. Finally, it selects samples from each cluster and uses them to augment M. This augmenting process, called FineTune, is left as a black box.

The second method is NearPrivate. The given motivation for NearPrivate is that the public dataset may have a different distribution than the original private training data. To address this, NearPrivate uses DP-PCA (a pre-existing technique for differentially private PCA) on the private dataset and projects both the private and public data onto the top k principal components. Then it compares uncertain points in the public and private dataset projections and only augments M (again using the black box FineTune) using uncertain public examples with many nearby uncertain private samples. Intuitively, this should select for points that are not too different than the original private training data.

The paper then evaluates these algorithms with experiments on MNIST and SVHN under reasonable privacy parameters (I think -- see below). These experiments compare the performance of DiversePublic and NearPrivate, where both train M using vanilla DP-SGD [Abadi et al. 2016]. DiversePublic and NearPrivate trade performances on MNIST and SVHN, but NearPrivate does (as might be expected) better when the public dataset is polluted.

In summary, the paper suggests DiversePublic and NearPrivate as useful ways to augment a given differentially private model M using public data, and their experiments suggest this is reasonable.

---DECISION---
Reject.

This is an empirical paper. It proposes some algorithms and justifies these algorithms through experiments. However, the paper makes a confusing omission: it does not mention PATE [1, 2].

[1] "Semi-supervised Knowledge Transfer for Deep Learning from Private Training Data". Papernot et al. 2017.
[2] "Scalable Learning with PATE". Papernot et al. 2018.

I am more of a pure differential privacy researcher and am therefore less familiar with this line of mostly empirical work. But as far as I can tell PATE works in the same setting as that of this paper: there is a private dataset, a public unlabelled dataset, and the goal is to train a model with a differential privacy guarantee for the private dataset. As a result, PATE seems directly relevant. In particular, the claim that "there are no published examples of a differentially private SVHN classifier with both reasonable accuracy and non-trivial privacy guarantees" seems wrong, as the PATE papers include just that.

Moreover, PATE appears at least competitive with the algorithms here. For example, the comparison given as Table 1 in [2] suggests that PATE can get 98-99% accuracy on MNIST for privacy parameters at least as low as those used here with only a few hundred new labelings. In contrast, if I understand the experiments here, they are using thousands of new labelled examples to get the same accuracy and privacy. Similarly, on SVHN the experiments here get to 84% accuracy with 10,000 new labelled examples, whereas the same Table 1 puts PATE at 90% with almost identical privacy parameters and thousands fewer new labelled examples.

That comparison is my main concern. Perhaps I am missing a reason why PATE is not comparable. If it is, it should certainly appear in the experiments. If PATE is indeed comparable, then the main improvement contributed by this paper is the post-processing aspect: PATE is a way to train a private model from scratch, but these methods work on an existing model. Even then, the question of how exactly the existing model is modified (the FineTune method) is unclear.

More broadly, there are several parts of this paper that could be much clearer. I am sympathetic to paper length limitations, but I am certain many or all of these issues can be addressed in 8 pages.

1. DiversePublic description: What does it mean to "obtain the 'embeddings'...E_{public}"? How is k picked?
2. How does the private point-public point assignment in NearPrivate work? Is it just random? How many points do we pick?
3. In general, how is FineTune meant to work? I understand it is meant to be an abstract black box in the algorithm descriptions, but the experiment description doesn't explain it either. How does FineTune work in the experiments?
4. What does 'headroom' mean in Section 4.2?
5. The way privacy parameters are specified at various points in the experiment section makes things hard to read. A concise table summarizing privacy parameters, accuracy, and additional labels would help.

But overall, I am most interested in the PATE comparison.

---EDIT AFTER AUTHOR RESPONSE---
I missed the data-dependent aspect of PATE's privacy guarantee. That is a point in favor of this paper. As such, I've increased my score from reject to weak reject.

I actually think a revised version of this paper could reasonably appear in ICLR or similar venues. However, I think the necessary revisions are too large/numerous to accept the paper in its current form, even with promises of revisions. Here are a few revisions that I think would make the paper a much better submission to future conferences:

1. Add discussion of PATE. The data-dependence of their privacy guarantees is well-taken. But it is not clear that data-dependent privacy guarantees are as "trivial" as the paper currently claims with "there are no published examples of a differentially private SVHN classifier with both reasonable accuracy and non-trivial privacy guarantees" suggests. (In fact, I find it confusing that one of the PATE authors agrees with this claim -- they think their own work is trivial?). I agree that data-independent privacy guarantees are preferable, but this should certainly be discussed. As the authors themselves note, the original PATE paper was well-received, so if this paper wants to claim its privacy guarantees are somehow meaningless or even just unsatisfactory, that claim needs to be defended.

2. Elaborate on the fine-tuning process. Perhaps "fine tuning" is indeed "super standard in the object recognition literature", but as all three reviews here indicate, the presentation of fine tuning is unclear in this paper. The author responses also seem to simultaneously claim that fine tuning is  both "super standard" and "[not] obvious".

3. To make room for the text above, it's not clear that the material about data pollution or even presenting both methods is necessary. I would prefer to see a thorough explanation of the active learning/post-processing paradigm through one algorithm. The authors seem to want to claim that active learning is a useful private approach because it gets data-independent privacy guarantees and performance similar to algorithms with data-dependent privacy guarantees. A paper that focuses on and thoroughly defends that claim actually sounds pretty good!

**Experience Assessment:**

I have published one or two papers in this area.

**Review Assessment: Checking Correctness Of Derivations And Theory:**

N/A

**Review Assessment: Checking Correctness Of Experiments:**

I assessed the sensibility of the experiments.

**Review Assessment: Thoroughness In Paper Reading:**

I read the paper thoroughly.

---

> ### Author Response · Authors · 2019-11-09
> **Thanks for the review!**
>
> We are working on a more thorough response, but in the meantime we  wanted to clarify:
>
> There is indeed a reason why PATE is not comparable:
> The bounds for PATE are data-dependent, unlike the bounds for DP-SGD.
> See Section 3.3 of [1] for more on this.
> Data-independent bounds for PATE would be much higher.
> We double checked this with an author of [2].
>
> We actually ran the claim "there are no published examples of a differentially private SVHN classifier with both reasonable accuracy and non-trivial privacy guarantees" by one of the PATE paper authors before submission and they were OK with it.
>
> Sorry for the confusion on this point, we should have been much more clear about it in the paper.
>
>
> [1] "Semi-supervised Knowledge Transfer for Deep Learning from Private Training Data". Papernot et al. 2017.
> [2] "Scalable Learning with PATE". Papernot et al. 2018.

---

> ### Author Response · Authors · 2019-11-15
> **Responses to your numbered questions:**
>
> Responses to your numbered questions:
>
> 1. These are just the activations for the second to last layer of the classifier.
> This terminology is pretty common in the object recognition literature.
> k is chosen with standard hyper-parameter optimization.
>
> 2. From the text:
> 'we assign each uncertain private example to exactly one uncertain public
> example according to Euclidean distance'
>
> 3. Sorry for the confusion here: 'Fine-tuning' as a term is also super standard in the object
> recognition literature: it means that you take a trained model and update its weights using
> new data. Generally, it's understood that you use a lower learning rate, and perhaps 'freeze'
> some of the weights of the earlier layers of the neural network.
>
> 4. Just that SVHN accuracy benchmarks are further from being saturated than MNIST ones.
>
> 5. Understood.

---

### Official Review · AnonReviewer5 · 2019-11-06
**Official Blind Review #5**

**Rating:** 3

**Review:**

This paper considers the problem of how to improve the performance of an existing DP classifier, with the help of the labelled public data. The paper considers the following process: 1. find a public dataset containing relevant samples; 2. carefully select a subset of public samples that can improve the performance of the classifier; 3 fine-tune the existing classifier on the selected samples. Two different techniques from active learning are utilized in order to select representative samples from a public dataset and fine-tune a DP classifier. This paper also conducts some experiments on   the MNIST and SVHN datasets and demonstrates improvement compared with the benchmark.

I vote for rejecting for this paper, because of the following two concerns:

1. I do not think this paper has made a lot of contribution to either differential privacy or active learning. It just "borrows" some fine-tuning techniques from active learning and apply it in DP. There is almost no theoretical contribution made by this paper. Besides, from the experimental perspective, neither can I see an obvious improvement compared with the benchmarks.

2. I do not think the privacy analysis of the NearPrivate algorithm (Algorithm 2) is correct. The paper uses private argmax algorithm and claims that it satisfies $eps_{support}$-DP. However, this is only true when $N_{labeled} = 1$. Generally, it should satisfy $eps_{support} \cdot N_{labeled}$-DP. So if we look at the experimental setting of MNIST, roughly thousand times less noise is added! Since this amount of noise is non-trivial at all, I can not judge the effectiveness of the algorithm.
------------------------------------------------------------------------------------------
Thanks for the authors' classification. I missed the part that each private sample was only assigned to one public sample. Now I can confirm the correctness of the algorithm and increase my score accordingly.

**Experience Assessment:**

I have published one or two papers in this area.

**Review Assessment: Checking Correctness Of Derivations And Theory:**

I assessed the sensibility of the derivations and theory.

**Review Assessment: Checking Correctness Of Experiments:**

I assessed the sensibility of the experiments.

**Review Assessment: Thoroughness In Paper Reading:**

I read the paper thoroughly.

---

> ### Author Response · Authors · 2019-11-15
> **Thanks for the review!**
>
> We wish to clarify why the sampling procedure of our NearPrivate method is differentially private with cost $\varepsilon_{support}$.
>
> For each uncertain private example, we assign it to one and only one uncertain public example according to Euclidean distance. For each uncertain public example, this yields a count $N_{support}$ of how many uncertain private examples are 'nearby'. This count $N_{support}$ has sensitivity of 1 with regard to each uncertain private example. We add LaplaceNoise(1/$\varepsilon_{support}$) to this count $N_{support}$, and sample from the uncertain public data according to the noisy $N_{support}$. This sampling procedure is differentially private with cost $\varepsilon_{support}$ due to the Laplace mechanism [1].
>
> [1] "The algorithmic foundations of differential privacy". Dwork and Roth. 2014.

---

> ### Author Response · Authors · 2019-11-15
> **Thanks for updating your review.**
>
> Thanks for updating your review regarding the bounds.
> For posterity, we respond to your other points:
>
> > I do not think this paper has made a lot of contribution to either differential privacy or active learning.
> Yes, we are not claiming to have made such contributions.
> What we claim is having presented a way to substantially reduce the gap between differentially private classifiers and normal classifiers:
>
> If you want to actually use differentially private classifiers in real life, you will be sad,
> because the accuracy will be poor.
> We point out that you can improve accuracy substantially by fine-tuning on public data after the fact.
> This is already something that was not in the literature.
> I don't think you can credibly claim that it's 'obvious' that fine-tuning on a totally different data-set
> after the fact will improve performance on your test set;
> this is not what anyone uses fine-tuning for currently.
> We also give two algorithms for doing this fine-tuning (both of which are new, by the way), and we give
> a privacy analysis for the algorithm that needs it.
>
> > There is almost no theoretical contribution made by this paper.
> Yes, this is not a theory paper.
> This is not a theory conference either.
>
> > Besides, from the experimental perspective, neither can I see an obvious improvement compared with the benchmarks.
> We increase the state-of-the-art for differentially private MNIST from 97.5% to 98.8%.
> Experimentally, that's a significant jump - it's more than a halving in error-rate.
> The improvement in SVHN is even more significant.
> Perhaps some of the confusion here is around benchmarks:
> Most of the benchmarks in the paper are in fact *benchmarks that we have set in this paper*.
> We claim that a paper that only pointed out how helpful fine-tuning can be for DP classifiers
> would, by itself, be worth publishing; we have written such a paper, and then added even more things.

---

### Decision · Program_Chairs · 2019-12-19

**Decision:**

Reject

**Comment:**

This paper provides an active-learning approach to improve the performance of an existing differentially private classifier with public labeled data. Where the paper provides a new approach, there is a consensus among the reviewers that the paper does not provide a strong enough contribution for acceptance. The authors can potentially improve the submission by including a more comprehensive comparison with the PATE framework and improving its overall presentation.